# Modeling Malaria Outbreaks Utilizing Weather Factors

## Abstract

Using meteorological data, time series forecasts of disease outbreaks can better
capture the true epidemiological profile of tropical diseases such as malaria. In this
study, several methods of time series analysis are employed to study the disease
patterns of malaria in the Indian state of Odisha. Weather information, including
temperature and precipitation data, is incorporated alongside monthly case numbers
in SARIMA and LSTM models. The viability of transferring the model trained on
malaria in Odisha to dengue in Bangkok, Thailand, is also explored. The methods
outlined in this paper can serve as the basis for forecasting mosquito-borne disease
outbreaks in settings with a poor data-collection infrastructure.

## 1 Introduction

One of the primary concerns of epidemiology is the prediction of disease incidence. From a public
health perspective, even approximate knowledge of the magnitude of future outbreaks can enable
healthcare systems to more effectively combat them. This is particularly true for systems in developing
countries, where the burden of disease and scarcity of resources are disproportionately high. As
the scale of data recording increases on a global level, we are better equipped than ever to employ
predictive modeling to forecast disease incidences in these regions.

Climate change is expected to radically impact global disease patterns. Extreme weather and natural
disasters all impact epidemic risk factors from vector distribution in tick-borne diseases to water
contamination illnesses like cholera. Consequently, many epidemiological papers link infectious
disease outbreaks with weather patterns. For example, climate factors such as temperature and
precipitation are suggested to impact the spread of malaria and other seasonal, mosquito-borne
diseases. [Thomson, 2005] Multiple studies have proven that machine learning algorithms that use
meteorological data combined with past outbreak data can reliably predict future local outbreaks
[Shaman, 2005] . Currently, researchers utilize neural networks as predictive models for seasonal
epidemics to decrease the burden on healthcare systems as well as eliminate risk factors that impact
outbreak severity. With the increased need to assess new epidemiological trends in light of climate
change, these seasonal forecast models will inform public health systems as to how future weather
patterns will influence disease outbreaks.

In this work, we aim to identify optimal prediction models for malaria incidence in the Indian state
of Odisha, which historically constitutes a large proportion of total cases in India. Among the 36
states and union territories of India, Odisha ranks 32nd on the Human Development Index. Due to
its geographic position, it is exposed to two wet seasons. This combination of socioeconomic and
environmental factors renders Odisha highly vulnerable to malaria outbreaks, and explains in part its
disproportionately large malaria burden. In developing our malaria incidence forecast models, we

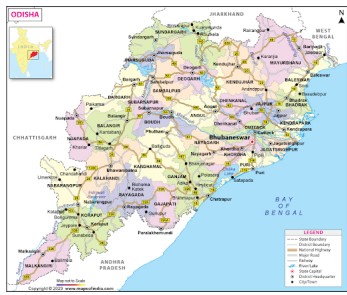

Figure 1: Map of Odisha and its districts

will investigate the effect of geospatial resolution at the district level on predictive accuracy, along with meteorological factors. Furthermore, we will explore the application of transfer learning to the problem of predicting mosquito borne outbreaks in other tropical regions.

## 2 Data

The malaria incidence data utilized in this paper comes from a study conducted by India's National Vector Borne Disease Control Program in the state of Odisha over the years 2003 to 2013. [thi]Provided are the monthly incidences of malaria in the 30 districts of the state.[fou] Time series were extracted from this study and serve as the basis for this paper. The meteorological data utilized comes from the TerraClimate dataset provided by the University of Idaho, which combines data from several sources with a degree of climatically aided interpolation. [fif] The dataset contains information on temperature, precipitation, vapor pressure, solar radiation, and wind across the globe from 1958 to 2020. TerraClimate was accessed using the Google Earth Engine API. Using the latitudes and longitudes of the 30 districts of Odisha, monthly precipitation, minimum temperature, and maximum temperature were extracted for the years 2003 to 2013 and stored as time series.

## 3 Methodology

### 3.1 SARIMA

SARIMA refers to the class of Autoregressive Integrated Moving Average Models (ARIMA) which are modified to account for seasonal variation in time series data. [Durbin and Koopman, 2012] ARIMA models are themselves derived from Autoregressive Moving Average Models (ARMA), and differ in that they are able to effectively characterize non-stationary data through integration. Non-stationary time series are those in which the probability distribution underlying the process changes over time. In the context of forecasting disease, non-stationarity presents itself through seasonality and an overall downward trend in cases over time, which can be attributed to public health interventions and socioeconomic shifts.

ARIMA models are defined by a set of three parameters (p, d, q), representing the orders of autoregression, differencing, and moving-average, respectively.

A basic ARMA model has the following form: $y_t = \mu + \sum_{i=1}^{p} \phi_i y_{t-i} + \sum_{i=1}^{q} \theta_i \omega_{t-i} + \omega_t$ The first summation represents the autoregressive portion of the model, where a linear combination of the previous p terms is used in determining the output. The second summation is a linear combination of the previous q white noise error terms associated with each element of the time series. ARIMA models deal with the issue of non-stationarity in the data by integration, the differencing of consecutive terms in the time series. This can be represented as $y'_t = y_t - y_{t-1}$, and the differencing operation can be performed multiple times (as dictated by the order of difference d) in order to accommodate time varying trends.

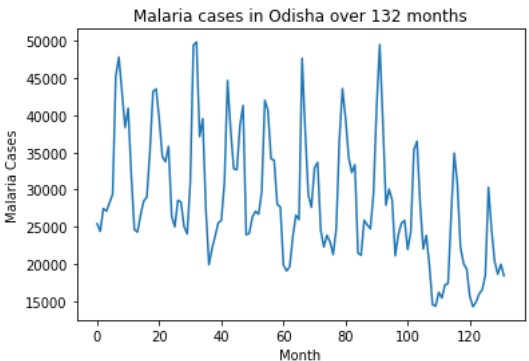

Figure 2: Seasonal Malaria Cases Aggregated Across all Districts in Odisha, 2003-2013

SARIMA models possess the additional advantage of being able to incorporate seasonal trends into the model via the inclusion of three additional terms (P,D,Q), which capture macro-level seasonal behavior in the time series.

## 3.2 SARIMA with Exogenous Regressors

In the context of the SARIMA objective functions described above, the incorporation of independent variables can be accomplished via their inclusion as weighted inputs to the forecast equation.

## 3.3 Note on VAR/VMA/VARMA

Vector Autoregression (VAR), Vector Moving Average (VMA), and Vector Autoregression Moving Average (VARMA) are members of a class of models which capture the linear relationship between multiple variables (i.e. a vector) over time [Hyndman and Athanasopoulos]. In this context, the modelling process assumes a degree of correlation between all of the quantities being modeled. The general class of models (VARMAX) is defined as so:

$$Y_t = \sum_{i=0}^{p} \phi_i Y_{t-i} + \sum_{i=0}^{b-1} B_i X_{t-i} + \sum_{i=0}^{q} \theta_i E_{t-i} + C + E_t$$

Where $Y_t$ is stationary endogenous variable, $\phi_i$ is the autoregressive component, $B_i$ are the exogenous regressors, $\theta_i$ is the moving average component, C is the vector constant, and $E_t$ is the residual error. Past lags of all variables in the system impact the forecast of all endogenous variables. As this research is concerned with the relationship between multiple variables and incidence of malaria, and not vice versa, the usage of these models was deemed inappropriate. [Malki, 2020]Indeed, while there may be a relationship between meteorological factors and malaria incidence, the same cannot be said of the reverse. Instead, a means of modeling a seasonal time series while incorporating additional independent variables was required, for which reason an SARIMA model with exogenous variables was employed.

## 3.4 LSTM

Long Short-Term Memory (LSTM) architectures are a variant of Recurrent Neural Networks (RNN) with feedback connections. The core difference between RNNs and traditional feedforward neural networks is the fact that RNNs have loops, allowing previous information, or a memory, to persist. This looped nature makes these models particularly well-suited for applications dealing with time series data. However, the performance of vanilla RNNs degrades when trying to integrate long-term dependencies into the model as a result of the so-called vanishing gradient problem which causes a neural network to stop training partway through. Therefore, a plain RNN is not well-suited for the study being conducted here. LSTMs combat the vanishing gradient problem through the inclusion of long-term memory in the form of cell state. Within each hidden state, the LSTM can add and subtract information from the cell state using forget and input gates. In this way, information can be

propagated across much longer sequences than in a typical RNN. For our application, where forecasts must take not only the annual cycle of malaria incidence into account, but also the overall downward trend in cases across many years, LSTMs were deemed well suited.

# 4 Methodology

In selecting the parameters for the SARIMA model, the autocorrelation and moving average components were chosen on the basis of which minimized the mean absolute error. The integration parameter was selected using autocorrelation (ACF) and partial autocorrelation (PACF) functions in order to determine which order of differencing maximized the stationarity of the data. ACF determines the correlations of present values with previous ones, while PACF determines the correlations of residuals. In transforming our data to a stationary time series, the goal was to find the order of differencing that minimizes ACF and PACF. A grid search of the parameter space was conducted using Auto ARIMA from the Pmdarima library in order to determine which parameters produced the most optimal results under these criteria.

In order to run our data through an LSTM, the data had to be reshaped into multidimensional arrays. Our inputs for $x_{train}$ were given an added third layer to establish proper dimensionality for an LSTM layer.

# 5 Results

## 5.1 SARIMA Baselines

Prior to constructing the LSTM model, a baseline SARIMA model was constructed. First an SARIMA was used to forecast aggregated malaria cases across Odisha. The endogenous variable was raw malaria cases, and the optimized parameters were order(2,1,1) and seasonal order(1,0,1,12). 100 iterations were ran with a random training set of 60 timepoints to forecast the next time step. Absolute error of the predicted and expected cases was calculated and averaged to evaluate the SARIMA's performance. Since this paper is interested in the impact of climate factors on malaria case load, an experiment was ran to test how SARIMA accounts for weather using case prediction. For each district in Odisha, a vanilla SARIMA and an SARIMA with exogenous regressors (precipitation, minimum temperature, maximum temperature, and peak rainfall per year for that particular district) were ran to compare how climate factors impact the SARIMA performance. 70 iterations with a training set of 60 time points used to forecast the malaria cases for the next time set were ran. Absolute error was once again calculated. See results folder in the code repository for list of hyperparameters, mean, standard deviation, and average absolute error for each model.

Overall, the performance for both SARIMA models highly vary across districts. Interestingly, four out of the five southern coastal districts (Kalahandi, Kandhamal, Keonjhar, Gajapati, and Ganjam) have the lowest median absolute error. The data suggest SARIMA's performance in each district is highly dependent on characteristics including geography and demographics.

Ultimately, the inclusion of exogenous weather factors do not improve the performance of the SARIMA model. Despite multiple studies showing the relationship between climate and seasonal disease outbreaks, the model shows poorer performance with the inclusion of extra data. Because SARIMA can only account for linear exogenous regressors, it fails to detect the complex relationships between climate and cases.

## 5.2 LSTM

In order to better be able to integrate the exogenous regressors into the model, an LSTM was used and run through a dataframe that was an aggregate of cases and weather predictions over time for all districts. Prior to being used in the model, the data was normalized and reshaped to be compatible with the LSTM and reasonable to make predictions from. The LSTM contained only two LSTM layers with 50 epochs of training (each taking  3s to run), and thus it has a significantly faster runtime

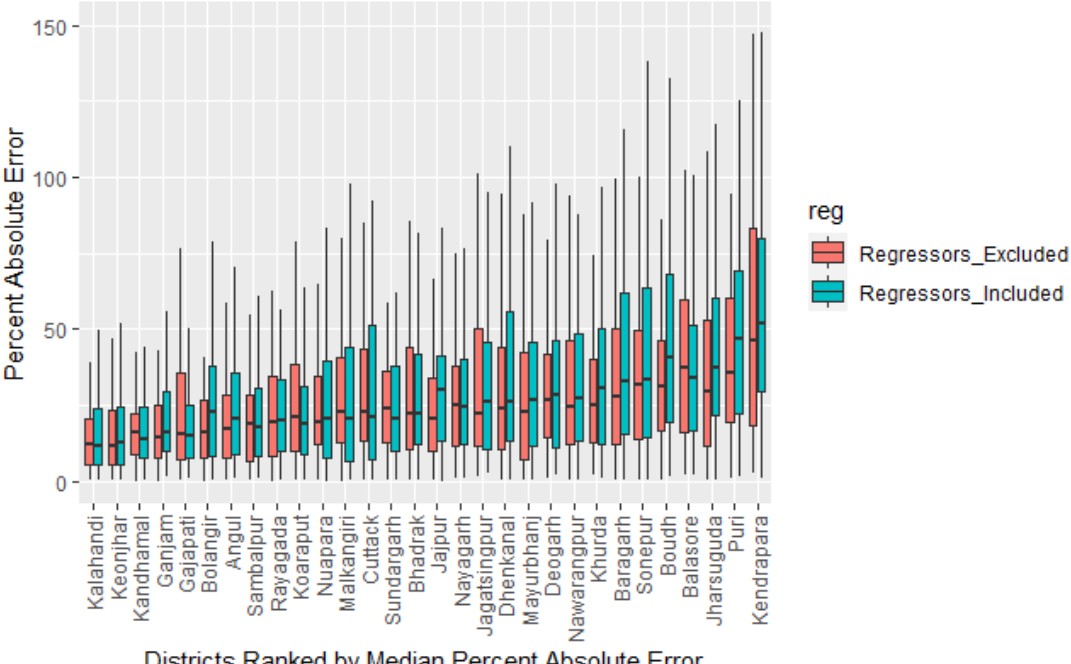

Figure 3: SARIMA Absolute Error with and without exogenous Regressors in Odisha Districts

Table 1: SARIMA Absolute Error Summary of Results

|  | Aggregated Over State | Per Districts | Per Districts + Regressors |
| --- | --- | --- | --- |
| Mean | 9.94 | 24.29 | 26.64 |
| Median | 8.59 | 32.162 | 36.25 |
| Standard Deviation | 7.50 | 32.68 | 36.48 |

of under 3 minutes compared to ARIMA or VAR models. This model resulted in a mean absolute error of 2.146% in cases predicted.

# 6  Transfer Learning Applications

Transfer learning is known to be effective for prediction on small timeseries datasets that would otherwise be insufficient. It has well known medical applications where data is particularly scarce.

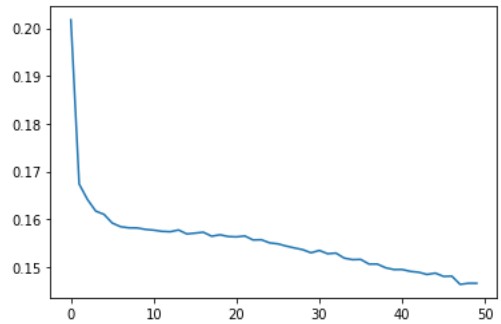

Figure 4: LSTM Loss Function

Due to limited funding and resources, many countries do not have access to detailed records on case numbers. This leads to smaller datasets that increase errors in forecasting. Because many tropical diseases, particularly those with mosquito vectors, follow similar seasonal trends, it's plausible transfer learning can increase case prediction accuracy on smaller dataset inputs. For this, a dengue dataset surveying the caseloads in Bangkok, Thailand from the years 2003 through 2017 was obtained. Historical average temperature (°C), humidity, and precipitation (mm) were also included as features.

To test this theory, the parameters from the LSTM trained on the Odisha dataset (2002-2013) were transferred to another LSTM used to predict a random two year subset of the Bangkok data. Since mosquito patterns vary depending on the climate, two experiments were performed: in one, parameters were transferred from an LSTM trained on a coastal Odisha district in a climate similar to that of Bangkok and in the other, parameters were transferred from an inland district [Polwiang, 2020]. These experiments aim to test how applicable transfer learning is between climates.

## 7 Discussion

With our processing and modelling pipeline, models with low levels of error were produced in an attempt to predict the number of active disease cases in an area. Both statistical models and machine learning models were attempted, and ultimately the machine learning models yielded the lowest levels of error in predicting results, with mean absolute error of <5 % on two datasets. This difference that we can see is likely because the LSTM network we utilized can better take into account the cyclical nature of weather patterns and these mosquito-borne disease outbreaks by keeping a memory of previous time steps. While transfer learning for other disease prediction is a future application of these models, it was not possible due to a lack of available data in many places. The team was, however, able to run the same model on data from a different tropical region (Bangkok, Thailand) with another mosquito-borne disease (dengue fever) and this resulted in an even lower error in case count of 0.64 %. Thus, this and similar models perform well where data is available. In the future, using weather data with a pre-trained model may allow for transfer learning on other diseases in different regions, which could allow countries to better allocate medical resources by forecasting disease outbreaks. This could work well for other tropical mosquito-borne diseases such as yellow fever, zika virus, and west nile virus.

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
