# OpenReview forum: "Modeling Malaria Outbreaks Utilizing Weather Factors"
_uoft.ai/University_of_Toronto/2021/ProjectX — Submitted to ProjectX2021_

### Official Review · Reviewer_tZuX · 2022-02-07
**Modeling Malaria Outbreaks Utilizing Weather Factors**

**Rating:** 4
**Confidence:** 5

**Review:**

**Connection to Current Science (science and practice)**

2

- There is sufficient connection to the field, but more depth on LSTM or ML applied to malaria detection could have easily been added
- Deployment details are not really touched on.

**Clarity of Communication**

1

- Figure 1 is so small as to be unreadable. It’s not clear what it adds to the paper.
- Some citations appear strangely in the text (e.g., in Sec 2)
- is the use of ‘not vice versa’ on line 86 meant to imply a causative approach and not a correlative one? The following discussion (lines 87-88) does not discount the possibility of correlation analysis.
- the paper flows well, with an easy-to-follow structure, and concise, clear explanations. The language is also easy to read.
- the invocation of ‘loops’ on line 94 does not really convey anything useful.
- Nothing is really said about the LSTM loss function (Fig 4)


**Methodological Quality**

2

- It’s not clear if a single point (presumably) of a latitude and longitude in a district is entirely sufficient to represent that district if there is a lot of variation, but this is perhaps a good enough simplification.
- This SARIMA model is a good choice, generally
- it’s not exactly true that the vanishing gradient problem ‘causes’ a neural network to stop training.
- the explanation of LSTMs are fairly superficial. You may prefer to simply refer to an LSTM paper rather than give an incomplete explanation.
- it’s not really clear how “the autocorrelation and moving average components were chosen…[to minimize] mean absolute error”. This actually makes reproducibility difficult, too.
- it’s not clear what ‘an added third layer’ refers to (line 116), nor how the LSTM and SARIMBA models were ‘trained’ on the same subsets of data (nor how those subsets were determined), in order to make validation/testing valid or comparable.


**Reproducibility**

0.5

- More should have been said about the datasets (e.g., number of data points, statistics on how they were distributed, etc)
- More should have been said about some of the parameters on line 61 in Sec 3.1. E.g.m what’s \mu? What’s \theta? How are they determined, etc?
- The parameter ‘d’ doesn’t appear anywhere before line 61?
- Again, some parameters of the equation on line 81 are left undefined.

---

### Official Review · Reviewer_mzUA · 2022-02-09
**Impressive start to a machine learning approach to predicting patterns in malaria using weather-related variables**

**Rating:** 6
**Confidence:** 3

**Review:**

Connection to current science: 2/3
* It would have been ideal to see some justification of the importance of malaria (e.g., statistics describing the burden of malaria in the population). As a reader, I would like to know the impact of malaria on factors such as morbidity, mortality, healthcare utilization, etc.
* More information is needed in the introduction to provide insight into how novel this paper is. For example, did any previous papers take a machine learning approach to using weather data to predict patterns in mosquito-borne diseases? If applicable, how does this paper build on those (i.e., address knowledge gaps and/or limitations)?
* The authors discussed potential applications and implications of their work, but did not describe pathways towards implementation/knowledge translation.

Clarity of communication: 1.5/2
* The paper is clear, easy to follow, and enjoyable to read.
* Some figures are very small with very small text, making them difficult to read
* The paper is presented in a logical structure

Methodological quality: 3/4
* The authors provided clear justifications for the use of their methods, while acknowledging some potential limitations.
* It is not clear what the authors mean by “many countries do not have access to detailed records on case numbers”. Does this suggest that many cases were missed due to limited testing capacity, data collection, or something else? If so, it would have been ideal to include some discussion of its implications on the results and/or application of the models.
* It would have been interesting to identify and report on which weather factors were the most influential predictors of patterns in malaria.
* It would have been ideal for the authors to indicate whether the methods used in the paper are new/novel and improve on existing work (and how/why).

Reproducibility: 1/1
* Impressive work considering this project was conducted over a period of 5 months.
* The authors indicate that the code has been placed in a repository for others to access and reproduce their work.

---

### Official Review · Reviewer_2oWM · 2022-02-10
**Well-written, could benefit from some additional experiments**

**Rating:** 6
**Confidence:** 3

**Review:**

### Summary of paper
The authors describe the use of SARIMA and LSTM models for forecasting malaria in an Indian state as a function of select climate factors, and explore whether the trained LSTM model can be used to forecast dengue in Bangkok.

### Connection to current science & practice
This paper shows good understanding of subject matter. Although similar applications of ARIMA and LSTM models have been used in the literature for forecasting infectious disease trends, which the authors could have discussed more, the authors implement important time series models assess transfer learning-based approach to prediction, which may be a novel application of such models. Perhaps the authors could've made clearer why models are trained at the level of individual districts within one Indian state - a pan-Indian model would be more generalizable to places around the world, either based on domain knowledge/literature or in terms of how it might've affected prediction. The authors provide a good overview of steps to implementation. 1.5/3

### Clarity & communication
The paper is well-written and flows well. It was not clear in the introduction that SARIMA and LSTM were being compared, I thought initially that they were proposing some sort of a hybrid model. Figures and tables should be referenced within the text and figure 4 could use axis labels. An enjoyable read overall. 1.5/2

### Methodological quality
Methods are well-reasoned and appropriate for the problem. Although no new methods or techniques are proposed, I think appropriate methods for time series analysis were used and unnecessary complexity was avoided generally for what the authors set out to do, although hyperparameter selection for LSTM could've been better justified. Perhaps evaluation metrics other than or in addition to mean absolute error could have been tested, or the authors may have benchmarked their results against prior published results for predictive models, e.g. for another infectious disease with similar etiology, or based on domain knowledge. Also, seems like the authors used 60 data points to predict only the next single time step? It might've been nice to test predictions over longer time steps. 2/4

### Reproducibility
The authors have made considerable efforts to be transparent in their approaches throughout the text and to allow others to reproduce their results. They also mention that they provide a code repository with hyperparameters and results. This seems reasonable work for a 5-month project. 1/1

---

### Official Review · Reviewer_YjqR · 2022-02-15
**A very good attempt at a challenging problem**

**Rating:** 6
**Confidence:** 3

**Review:**

1. Connection to Current Science 1 out 3

The problem was well motivated but I struggled to be convinced that the relevant literature was explored enough. I am not saying this should have been exhaustive (as it is a large area) but I got the impression that there was a lack of positioning of the work within the broader set of applications. For example, COVID with a changing incidence could have been cited as remotely relevant to this and from there ideas for modelling and predicting could have been added.

2. Clarity of Communication 2 out of 3

The paper was mostly well written and the authors were very clear in explaining the process of thinking they had followed. Figures were very well used.

3.Methodological Quality 3 out of 4
To the best of my knowledge the work done seems very good for the time they had given. The comparisons seem relevant and well grounded. Results seem plausible and well explained. There are some gaps (e.g. in the limited selecting which methods to compare to) but overall I think the authors did a very good job and hard work went into this.

4.Reproducibility 0 out of 1
I did not find codes to check results (but I did not spend time looking for this either). I found the paper clear enough to follow.

---

### Official Review · Reviewer_rEtC · 2022-02-16
**Overall the paper describing modeling of malaria outbreaks utilizing weather factors is well written and the methodology is well described but the approach lacks originality and innovation and the paper does not describe how their models compare in performance to other similar models that have been developed for this purpose**

**Rating:** 7
**Confidence:** 3

**Review:**

Overview
- Time series analysis to study disease patterns of malaria in the Indian state of Odisha
- Temperature and precipitation data
- SARIMA and LSTM models
- autocorrelation and moving average parameters tuned based on mean absolute error
-integration parameter selected using autocorrelation and partial autocorrelation
- Generalizability of model trained on malaria in Odisha data to dengue in Bangkok, Thailand explored

Quality
- Well described, manuscript reads very well
- Able to access large amount of data - climate data was accessed via the Google Earth Engine API - TerraClimate 1958-2020 (monthly precipitation, minimum temperature and maximum temperature)  - and malaria data from India's National Vector Borne Disease Control Program in Odisha 2003-2013 (monthly incidence of malaria in 30 districts)

Clarity
- Methodology and approach are thoroughly described
- Unclear whether a mean absolute error of <5 % is acceptable in this context and how this compares to other models of this type that were developed to predict outbreaks
- Unclear whether monthly precipitation, min and max temp and monthly malaria data provides best prediction - would addition of other features improve prediction? what was the feature selection process -  how did you decide to include just historical malaria data, precipitation and temp in the models?
- SARIMA model performance varied by district within Odisha - authors assume due to differences in geography and demographics and inability of model to handle exogenous weather factors
- the LSTM model had a mean absolute error of 2.146%; performance by district was not described, although model trained on Odisha data for malaria outbreak prediction was tested on dengue data from Bangkok with good performance
- No clear summary/conclusion on which is the better model, in which circumstances or why

Originality
- Not very original, SARIMA models are often used to analyzed time series data
- They are widely used to detect outbreaks
- Usually used to predict short-term fluctuations of infectious diseases
- Other studies published the past decade have considered the use of ARIMA and other models to in malaria prediction/forecasting (ex: https://journals.plos.org/plosone/article/file?id=10.1371/journal.pone.0226910&type=printable)
- Novelty of approach is not described

Significance
- climate change, overcrowding, etc likely to lead to more frequent outbreaks
- models that can predict an outbreak using monitoring data that is collected at regular intervals would constitute a significant contribution to public health and safety

---

### Decision · Program_Chairs · 2022-02-19

NA